# Association between Masticatory Performance, Nutritional Intake, and Frailty in Japanese Older Adults

**DOI:** 10.3390/nu15245075

**Published:** 2023-12-12

**Authors:** Mitsuzumi Okada, Yohei Hama, Ryota Futatsuya, Yoshiyuki Sasaki, Kanako Noritake, Kohei Yamaguchi, Mayuko Matsuzaki, Chieko Kubota, Akemi Hosoda, Shunsuke Minakuchi

**Affiliations:** 1Department of Gerodontology and Oral Rehabilitation, Graduate School of Medical and Dental Sciences, Tokyo Medical and Dental University, Tokyo 113-8549, Japan; m.okada.gerd@tmd.ac.jp (M.O.); r.futatsuya.gerd@tmd.ac.jp (R.F.); ko.yamaguchi.gerd@tmd.ac.jp (K.Y.); matsuzaki.mayuko@tmd.ac.jp (M.M.); s.minakuchi.gerd@tmd.ac.jp (S.M.); 2Clinical Dental Research Promotion Unit, Faculty of Dentistry, Tokyo Medical and Dental University, Tokyo 113-8549, Japan; sasaki.prev@tmd.ac.jp; 3Oral Diagnosis and General Dentistry, Tokyo Medical and Dental University Hospital, Tokyo 113-8549, Japan; noritake.irm@tmd.ac.jp; 4Department of Oral Health Sciences, Meikai University, Chiba 279-8550, Japan; c.kubota@meikai.ac.jp; 5Division of Medical Nutrition, Faculty of Healthcare, Tokyo Healthcare University, Tokyo 141-8648, Japan; a-hosoda@thcu.ac.jp

**Keywords:** older adults, aged, masticatory performance, frailty, nutritional intake, long-term care prevention, protein intake, vitamin D intake, cross-sectional study, Japanese

## Abstract

The older adult population in Japan is expected to increase. Therefore, long-term care and frailty prevention are important. However, the relationship between masticatory performance, nutritional intake, and frailty remains unclear. This cross-sectional study aimed to examine energy, protein, and vitamin D intake and its association with frailty and masticatory performance in older adults. Patients between January 2022 and January 2023 were recruited and divided into robust and frail groups. Masticatory performance, nutrition, frailty, and other data, such as age and sex, were evaluated through onsite measurements and a questionnaire. Logistic regression analysis was conducted with frailty as a dependent variable and masticatory performance as an independent variable, adjusting for age, sex, skeletal muscle mass, living alone, energy intake, protein–energy ratio, and vitamin D intake. No significant differences were observed between the groups regarding age or sex. The robust group showed significantly better results for protein–energy ratio, vitamin D intake, and subjective and objective masticatory performance than the frail group. Logistic regression analysis revealed a significant correlation between skeletal muscle mass, protein–energy ratio, and objective masticatory performance with frailty. Masticatory performance was associated with frailty, independent of the intake of nutrients such as energy, protein, and vitamin D.

## 1. Introduction

Since 2007, Japan has entered a phase of being a super-aged society. This demographic transformation affects the future of the nation in various aspects, and this concern needs to be addressed. Moreover, the older population is expected to increase, and the aging rate will exceed approximately 38% in 2065 [1]. This demographic shift towards an increasingly aged population has significant implications for healthcare and societal well-being. In a super-aged society, the average life expectancy is increased; however, a 10-year gap exists between average and healthy life expectancy [1]. This implies that many older people require long-term care approximately 10 years before the end of life. An increase in the number of older people requiring long-term care not only lowers individual quality of life but also leads to an increase in medical and long-term care costs. This would further result in a decline in national power. As a result, long-term care prevention is a pressing issue in Japan that must be managed.

It is essential to prevent frailty before it leads to the need for long-term care. [2]. Frailty is a condition in which vulnerability to health disorders increases due to aging and there is a decline in reserve capacity. This concept was proposed by the Japan Geriatrics Society in 2014 [3]. Frailty is a multidimensional concept that includes not only physical frailty, such as weight loss and muscle weakness, but also mental and social frailty, such as decreased energy and activity. Furthermore, it has been reported that older adults are particularly susceptible to developing frailty. Frailty is considered to be located between a healthy state and a state requiring nursing care that requires support in daily life, and it is thought that recovery from frailty is possible. It has been reported that frailty increases mortality rates and causes a decrease in physical ability, so early detection and early intervention of frailty is extremely important for nursing care prevention [4].

The association between frailty and nutrition has been well documented in the literature. Reviews and meta-analyses have reported that frailty prevention requires sufficient energy and protein intake [5,6,7]. A study by Coelho-Junior et al. reported that protein-related parameters are associated with frailty and that frailty prevention requires more protein than the amount that is recommended in the dietary reference intakes established by the United States National Academies of Sciences, Engineering, and Medicine [5]. Moreover, they reported, in a systematic review and meta-analysis, that it is essential to increase protein intake in older adults to prevent the onset of frailty [6]. Another nutritional component that has come into focus is vitamin D. A decrease in serum vitamin D may increase the risk of falls, osteoporosis, and the need for long-term care [4,8,9]. This is possible because vitamin D deficiency is associated with loss of muscle strength and impaired bone turnover. Furthermore, consensus articles on frailty have mentioned that protein, energy, and vitamin D intake may prevent frailty [7] and that appropriate intake of these nutrients may be needed.

Several cross-sectional studies on independent, community-dwelling older adults have reported that masticatory performance is associated with frailty [10,11] and sarcopenia [12,13]. Iwasaki et al. evaluated the masticatory performance of 141 Thai older adults (mean age 72 years) and performed multivariate analysis to explore the association with frailty. Consequently, they reported that masticatory performance was a significant factor in frailty when analyzed using ordinal logistic regression analysis [10]. Murakami et al. investigated the association between masticatory performance and sarcopenia in a cross-sectional study of 761 Japanese older adults (mean age 73 years). As a result, they reported an association between masticatory performance and sarcopenia in logistic regression analysis [12]. Longitudinal studies have also reported that masticatory performance is associated with frailty and the need for long-term care [14,15,16,17]. In a systematic review of longitudinal studies of the association between oral health and frailty, Hakeem et al. reported that masticatory function is associated with the onset of frailty [14]. Similarly, in a longitudinal study of 2011 independent Japanese older adults (65 years of age and above), Tanaka et al. reported that masticatory function was associated with factors involved in long-term care needs, such as frailty and sarcopenia, as a result of analysis with the Cox proportional hazards model [17].

Furthermore, an association has also been reported between masticatory performance and nutritional status. Studies on the relationship between masticatory performance and nutrition have reported that decreased masticatory performance is associated with malnutrition, as assessed by screening tools and serum albumin levels [18,19].

As mentioned above, the relationship between nutrition and frailty, masticatory performance and frailty, and masticatory performance and nutrition has been reported in previous studies. Hence, various factors are involved in the association between masticatory performance and frailty, with decreased nutritional intake possibly contributing as a mediating factor. Therefore, it can be hypothesized that a decrease in masticatory performance leads to a decrease in nutritional intake, resulting in frailty. However, few studies have comprehensively examined the relationship between masticatory performance, nutritional intake, and frailty; hence, this relationship is unclear. It is meaningful to evaluate the relationship comprehensively among these three items at the same time and to examine their relationship to each other in order to prevent frailty by improving mastication and nutrition.

The purpose of this study was to examine energy, protein, and vitamin D intake, which are thought to be associated with frailty, to clarify their association with frailty and masticatory performance in older adults.

## 2. Materials and Methods

### 2.1. Study Design

This was a cross-sectional study.

### 2.2. Participants

Patients who visited the Tokyo Medical and Dental University Hospital were recruited between January 2022 and January 2023. The recruitment goal was 200 people as the maximum number of participants who could be assessed in 1 year. The eligibility criteria were as follows: patients aged 65 years or older, those who were not certified as needing support or long-term care, and those who were not currently undergoing dental treatment, except maintenance treatments such as oral cleaning and denture adjustment. The exclusion criteria were as follows: patients with systemic diseases such as cerebral infarction and orthopedic disease, those subject to dietary restrictions by specialists, and those with diagnosed or suspected dementia.

This study was conducted in accordance with the Declaration of Helsinki and was approved by the institutional review board of Tokyo Medical and Dental University (Approval number: D2021-043). All participants provided written informed consent.

### 2.3. Measurement/Questionnaire

Evaluation of frailty, objective masticatory performance, and other data were performed in a day. Questionnaires to evaluate nutrition, subjective masticatory performance, age, sex, and other data were distributed in advance, and participants were asked to answer them before coming onsite for measurements. Furthermore, the researchers confirmed that none of the questions in the questionnaire were unanswered to ensure that there was no missing data.

#### 2.3.1. Masticatory Performance

Masticatory performance was evaluated both objectively and subjectively. In this study, the objective masticatory performance was evaluated using gummy jelly. After the participant chewed 2 g of the gummy jelly freely for 20 s, they gargled with 10 mL of water. Following this, the gummy and water were discharged into a filtration mesh. The glucose concentration of water passing through the mesh was then measured with a masticatory performance test system (Gulco Sensor GS-II, GC Corp., Tokyo, Japan). The concentration of glucose correlates with the surface area of the chewed gummy jelly and is used as an evaluation value for objective masticatory performance [20]. Subjective masticatory performance was evaluated using question 13 from a dedicated questionnaire known as the Kihon Checklist. The question was as follows: “Do you have any difficulties eating tough foods compared to 6 months ago?” [21,22].

#### 2.3.2. Nutrients

Energy, protein, and vitamin D intake were evaluated using the brief-type self-administered diet history questionnaire (BDHQ) [23,24]. The BDHQ is a questionnaire that includes questions about the frequency of food and beverage intake in the past month and enables the estimation of the intake of nutrients based on the standard Japanese food composition table. The BDHQ was used to calculate the energy intake, protein–energy ratio (protein to daily energy intake), and vitamin D intake. The protein–energy ratio was calculated using the following equation:Protein–energy ratio: protein (g) × 4 (kcal/g)/energy intake amount (kcal) × 100(1)

#### 2.3.3. Frailty

Frailty was evaluated using the Japanese version of the Cardiovascular Health Study criteria (J-CHS criteria) [25]. Weight loss, muscle weakness, fatigue, walking speed, and physical activity were evaluated. Participants who did not fall under any of the following items were evaluated as “robust,” and those who fell under one of the items were evaluated as “pre-frail.” Those who fell under two or more items were evaluated as “frail.” The participants were classified into two groups. Participants evaluated as robust were classified in the robust group, and those evaluated as pre-frail or frail were classified in the frail group.

Weight loss: Response of “yes” to “Have you unintentionally lost two or more kg in the past six months?”

Muscle weakness: Grip strength, measured using a grip strength meter (Grip D, Takei Scientific Instruments Co., Ltd., Niigata, Japan), was <28 kg in men and <18 kg in women.

Walking speed: Measurement of the time taken to walk 6 m with a 1 m runway, resulting in a walking speed of 1 m/s or less.

Fatigue: Response of “yes” to “In the past 2 weeks, have you felt tired without a reason?”

Physical activity: Response of “not even once a week” to either question of “Do you engage in moderate levels of physical exercise or sports aimed at health?” or “Do you engage in low levels of physical exercise aimed at health?”

#### 2.3.4. Other Measurements

Skeletal muscle mass (skeletal muscle index) was measured using bioelectrical impedance analysis using a body composition analyzer (InBody 270, InBody, Tokyo, Japan). The age, sex, and type of residence (living alone, living together) of the participants were evaluated using the questionnaire.

### 2.4. Analysis

For continuous variables (age, skeletal muscle mass, energy intake, protein–energy ratio, vitamin D intake, and objective masticatory performance), receiver operating characteristic curves were drawn with frailty as the dependent variable and each continuous variable as the independent variable. A cut-off value, determined by the Youden Index, was used to convert binary variables. Significant differences in each measurement item between groups were determined using the chi-square test. The frail or robust groups were used as dependent variables, and objective and subjective masticatory performance and the other factors were set as independent variables. A logistic regression analysis was conducted with frailty as a dependent variable (reference: robust) and masticatory performance as an independent variable, with adjustments for age, sex, skeletal muscle mass, living alone, energy intake, protein–energy ratio, and vitamin D intake. The variance inflation factors (VIFs) were calculated to consider the multicollinearity.

All statistical analyses were conducted using JMP Pro16 (SAS Institute Inc., Cary, NC, USA). A *p*-value of <0.05 indicated statistical significance.

## 3. Results

A total of 213 people were recruited, and 200 participants (86 men and 114 women) were assessed. Based on the exclusion criteria, 18 participants (needing support: 1 person; diet control: 8 people; dementia: 9 people) were excluded, thus resulting in 182 individuals for analysis (Figure 1).

Table 1 shows the characteristics of the participants included in this study. There were 93 participants (51.1%) in the robust group and 89 (48.9%: 83 pre-frail and 6 frail people) in the frail group. The median age was 74.0 years (interquartile range (IQR): 71.0–77.5 years) in the robust group and 74.0 years (IQR: 69.0–80.0 years) in the frail group. The division of sexes in the groups was 38 men (40.9%) and 55 women (59.1%) in the robust group and 40 men (45.0%) and 49 women (55.0%) in the frail group. No significant differences between the groups regarding age or sex were shown. Protein–energy ratio was significantly higher in the robust group. Further, vitamin D intake was significantly higher in the robust group, although energy intake showed no significant difference between the robust group and the frailty group. Both subjective masticatory performance and objective masticatory performance were significantly better in the robust group. Living alone and skeletal muscle mass were not significantly different between the robust group and the frailty group (*p* < 0.05).

Table 2 shows the results of the logistic regression analysis. In the results, skeletal muscle mass (odds ratio (OR): 2.50), protein–energy ratio (OR: 2.47), and objective masticatory performance (OR: 2.06) had significant and independent correlations with frailty after adjusting for covariates (age, sex, living alone, energy intake, vitamin D intake, subjective masticatory performance). The maximum value of VIFs was 1.58.

## 4. Discussion

To the best of our knowledge, this is the first study to comprehensively examine the association between masticatory performance, frailty, and nutritional intake. The logistic regression analysis, after adjusting for multiple factors, showed that masticatory performance was associated with frailty, independent of nutrient intake.

Table 1 shows the characteristics of the study participants included in this study. A total of 89 participants (49%) met one or more J-CHS criteria and were classified as frail. The median value for the objective masticatory performance (160.5 mg/dL) of the study participants exceeded the test standard value for oral hypofunction proposed by the Japanese Society of Gerodontology [26]. The median energy intake was 1818 kcal, which is lower than the estimated energy requirement in the Dietary Reference Intakes for Japanese (2020 version) [27]; however, there was no extreme nutritional deficit. The average protein–energy ratio (17.3%) and the median vitamin D intake (17.0 μg) were within the target range and above the recommended amount in the Dietary Reference Intakes for Japanese (2020 version). Hence, the study participants had generally good masticatory performance and nutritional intake. Furthermore, they had completed dental treatments and were visiting a dental university hospital for maintenance treatment; hence, they might have been a group with a higher degree of health consciousness than the general population. Therefore, it should be noted that populations with other characteristics may have different results.

In this study, the analyses were adjusted with not only the basic information of age and sex but also confounding factors possibly associated with frailty, such as skeletal muscle mass [28] and type of residence [29]. The type of residence, in particular, was captured by differentiating between living alone and living with others. For nutrition, which was hypothesized to mediate the association between masticatory performance and frailty, adjustments were made for energy, protein, and vitamin D intake, which were thought to be associated with frailty as the confounding factors [5,8]. The target protein intake is set by the protein–energy ratio in the Dietary Reference Intakes for Japanese; hence, the protein-energy ratio was also used in this study. This ratio represents the proportion of energy derived from protein to total energy and is an indicator of nutritional balance.

Additionally, the measured values in this study were associated with frailty, but the dose–response relationship was poor; hence, these were categorized for analysis. Subsequently, there were 89 people in the frailty group, which had a lower incidence among the two outcome groups; in this case, we can use 9 independent variables for the logistic regression. Hence, because the number of independent variables is counted as 1 for binary variables, it was decided that they would all be converted to binary variables so that the number of independent variables would be within nine [30].

Similar to the findings of previous research, the protein–energy ratio, vitamin D intake, and subjective and objective masticatory performance were significantly higher in the robust group than in the frail group [10,15,31,32]. These findings are all believed to be reasonable results.

VIF is an index for evaluating multicollinearity, meaning the association among independent variables, and a value of 10 or more is considered a strong association between variables, which means inappropriate statistical analyses. The maximum value of VIF in the logistic regression analysis of this study was 1.58, which was not large, and it was determined that there was no effect of multicollinearity.

Before the start of the study, a hypothesis was established that decreased masticatory performance would lead to decreased nutritional intake, thereby resulting in frailty. However, we found that objective masticatory performance was associated with frailty independent of nutritional intake, as evaluated by the BDHQ. This indicates that factors other than nutritional intake evaluated by the BDHQ mediated the association between masticatory performance and frailty. When considering the relationship between nutritional intake and frailty in more detail, it was thought that the nutrients contained in ingested food are absorbed and reflected in the nutritional status, which is thought to be associated with frailty. In this study, nutrients were evaluated from food intake frequency using the BDHQ; however, the BDHQ did not evaluate the nutritional status, such as whether the ingested nutrients were absorbed. Some previous studies reported that ingested food is thought to be absorbed through oral processing, such as mastication [33,34,35]. Our study results could be explained by the fact that even if nutrients are ingested as food, low masticatory performance may result in inadequate absorption and insufficient improvement in nutritional status, resulting in frailty. However, systemic factors such as the presence of inflammatory and infectious states and malabsorption syndromes are also associated with nutrient absorption. Further research on this relationship requires evaluating nutritional status, such as blood tests, and these systemic conditions.

Subjective masticatory performance was not found to be a significant factor in frailty in multivariate analysis. In this study, subjective masticatory performance was the participant’s response to the question, “Do you have any difficulties eating tough foods compared to six months ago?” and represents whether participants themselves felt they had trouble chewing. It has been reported that subjective and objective masticatory performances are not necessarily associated with one another [36]. The fact that the BDHQ is a self-administered questionnaire may explain why subjective masticatory performance was more weakly associated with nutritional intake according to the BDHQ than objective masticatory performance in the present study.

This study has a few limitations that should be noted. First, this was a cross-sectional study. The causal relationship between masticatory performance and frailty remains unknown. Even if the decline in masticatory performance is a cause of frailty, it is unlikely that frailty develops immediately upon the decline in masticatory performance. Instead, it is more likely that it takes some time for masticatory performance decline to affect frailty. Thus, a long-term longitudinal study is needed to clarify the detailed associations. Second, though the use of gummy jelly to assess objective masticatory performance can aid in the evaluation of the mastication when measuring, it may not reflect eating behavior during daily mastication. However, daily mastication may have more substantial effects on daily nutritional intake. Therefore, it may be more useful if examinations that included eating behavior were conducted, as follows: measuring mastication parameters such as the number of chewing strokes or mastication speed using a dedicated device [37]. Third, a prior study found that the BDHQ has satisfactory ranking abilities, despite the fact that it only estimates mean values for a small number of nutrients [23]. The nutritional evaluation was converted to a binary variable for statistical analysis in our study, rather than utilizing mean values. As a result, we believe that there was no statistical problem. Furthermore, it should be recognized that for clinical applications, more detailed, specific nutrition assessment tools should be employed. Finally, frailty is thought to be a multifaceted phenomenon that includes not only physical aspects but also social and psychological ones [38,39], with decreased social participation believed to be a factor in physical frailty [40]. Masticatory difficulty is associated with the frequency of socializing [41]. In this study, the association between masticatory performance and physical frailty was discussed from the perspective of nutritional intake; however, future studies examining the relationship between masticatory performance and frailty should also evaluate social participation.

## 5. Conclusions

In this study, we comprehensively investigated the associations between masticatory performance, nutritional intake, and frailty. We found that masticatory performance was associated with frailty, independent of the intake of nutrients such as energy, protein, and vitamin D evaluated using BDHQ. In the future, a long-term longitudinal study that includes evaluations of eating behavior and social participation is needed to investigate these relationships in more detail.

## Figures and Tables

**Figure 1 nutrients-15-05075-f001:**
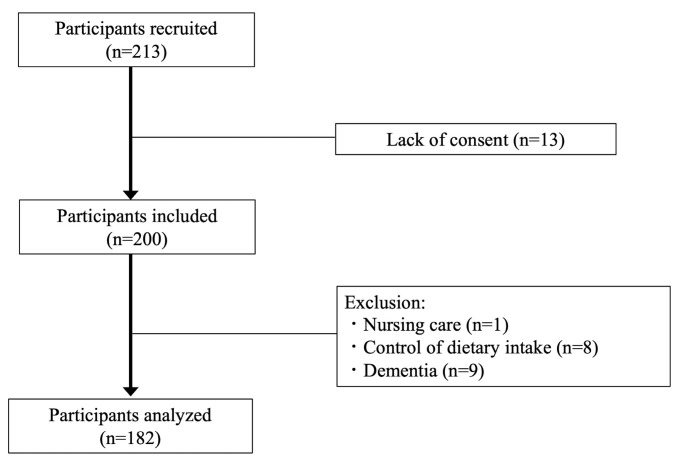
Participants flow chart.

**Table 1 nutrients-15-05075-t001:** Participant characteristics.

*n*		All (*n* = 182)	Robust Group (*n* = 93)	Frailty Group (*n* = 89)	*p*-Value
Age (years)		**74.0 (70.0, 78.3)**	**74.0 (71.0, 77.5)**	**74.0 (69.0, 80.0)**	
	Younger (≦79)	144 (79.1%)	78 (83.9%)	66 (74.2%)	
	Older (> 79)	38 (20.9%)	15 (16.1%)	23 (25.8%)	0.11
Sex					
	Women	104 (57.1%)	55 (59.1%)	49 (55.0%)	
	Men	78 (42.9%)	38 (40.9%)	40 (45.0%)	0.58
Living alone					
	Living together	149 (81.9%)	14 (15.1%)	19 (21.3%)	
	Living alone	33 (18.1%)	79 (84.9%)	70 (78.7%)	0.27
SMI (kg/m^2^)		**6.3 (5.6, 7.3)**	**6.2 (5.8, 7.2)**	**6.3 (5.5, 7.4)**	
	High (> 5.7)	126 (69.2%)	70 (75.3%)	56 (62.9%)	
	Low (≦ 5.7)	56 (30.8%)	23 (24.7%)	33 (37.1%)	0.07
Energy (kcal)		**1818.0 (1488.0, 2186.4)**	**1817.6 (1526.5, 2213.1)**	**1824.3 (1466.9, 2170.6)**	
	High (> 1547.3)	127 (69.8%)	69 (74.2%)	58 (65.2%)	
	Low (≦ 1547.3)	55 (30.2%)	24 (25.8%)	31 (34.8%)	0.18
Protein (% energy)		17.3 (16.9–17.8)	18.0 (17.4–18.7)	16.6 (15.9–17.3)	
	High (> 19.4)	46 (25.3%)	32 (34.4%)	14 (15.7%)	
	Low (≦ 19.4)	136 (74.7%)	61 (65.6%)	75 (84.3%)	0.003 *
Vitamin D (μg)		**17.0 (10.1, 25.7)**	**19.1 (10.5, 27.3)**	**13.6 (8.9, 24.7)**	
	High (> 9.5)	144 (79.1%)	81 (87.1%)	63 (70.8%)	
	Low (≦ 9.5)	38 (20.9%)	12 (12.9%)	26 (29.2%)	0.006 *
Subjective masticatory performance					
	No (good)	156 (85.7%)	85 (91.4%)	71 (79.8%)	
	Yes (bad)	26 (14.3%)	8 (8.6%)	18 (20.2%)	0.02 *
Objective masticatory performance (mg/dL)		**160.5 (121.0, 203.2)**	**171.0 (127.0, 217.0)**	**150.0 (116.5, 195.5)**	
	High (> 155)	95 (52.2%)	58 (62.4%)	37 (41.6%)	
	Low (≦ 155)	87 (47.8%)	35 (37.6%)	52 (58.4%)	0.005 *

Data are presented as average values with 95% confidence intervals (normal type) or median values with 25th and 75th percentiles (bold type) and the ratio of binarizations. *p*-values were calculated using the chi-square test. Abbreviations: SMI, skeletal muscle index. * *p* < 0.05.

**Table 2 nutrients-15-05075-t002:** Logistic regression analysis using frailty or robustness as a dependent variable (robust: reference).

		OR (95% CI)	*p*-Values	VIF
Age (years)				
	Younger (≦79)	1		
	Older (>79)	1.45 [0.62–3.36]	0.39	1.13
Sex				
	Women	1		
	Men	1.92 [0.84–4.42]	0.12	1.59
Living alone				
	Living together	1		
	Living alone	1.40 [0.61–3.25]	0.40	1.05
SMI (kg/m^2^)				
	High (>5.7)	1		
	Low (≦5.7)	2.50 [1.07–5.88]	0.03 *	1.46
Energy (kcal)				
	High (>1547.3)	1		
	Low (≦1547.3)	1.63 [0.76–3.50]	0.21	1.18
Protein (% energy)				
	High (>19.4)	1		
	Low (≦19.4)	2.47 [1.10–5.55]	0.02 *	1.15
Vitamin D (μg)				
	High (>9.5)	1		
	Low (≦9.5)	2.11 [0.90–4.96]	0.08	1.16
Subjective masticatory performance				
	No (good)	1		
	Yes (bad)	2.40 [0.90–6.44]	0.08	1.07
Objective masticatory performance (mg/dL)				
	High (>155)	1		
	Low (≦155)	2.06 [1.07–3.97]	0.03 *	1.07

Abbreviations: CI, confidence interval; OR, odds ratio; VIF, variance inflation factor; SMI, skeletal muscle index. * *p* < 0.05.

## Data Availability

Derived data supporting the findings of this study are available from the corresponding author, Yohei Hama, upon reasonable request.

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
