# Peer review of "Association between Masticatory Performance, Nutritional Intake, and Frailty in Japanese Older Adults"

_nutrients, 2023, doi:10.3390/nu15245075_

Round 1

Reviewer 1 Report

Comments and Suggestions for Authors

 Association between masticatory performance, nutritional intake, and frailty in Japanese older adults

The purpose of  study “Association between masticatory performance, nutritional intake, and frailty in Japanese older adults” was to clarify the relationship between masticatory performance, nutritional  intake, and frailty.

In truth, the relationship between these three parameters is evident in the scientific works reported by the authors in the introduction and the work does not provide further demonstrations or clarifications.

In particular, the work is lacking in the methods used for the evaluation of nutritional intakes and in the evaluation of the clinical and nutritional status of the subjects examined

Indeed, the evaluation of nutritional intake were evaluated using the brief-type self-ad- ministered diet history questionnaire (BDHQ)  based on a tool suitable for epidemiological studies, but not for clinical studies. (Both comprehensive  and brief self-administered diet history questionnaires satisfactorily rank nutrient intakes in Japanese adults. J Epidemiol 2012)

Furthermore, the study does not report parameters necessary for the clinical and nutritional evaluation of the subjects examined: body weight, BMI, kidney or liver pathologies, metabolic pathologies, presence of inflammatory or infectious states, malabsorption syndromes, intake of drugs or supplements. Furthermore, the blood value of vitamin D is not reported.

I therefore believe that the study is not scientifically valid and that the conclusions:  “We found that masticatory performance was  associated with frailty, independent of the intake of nutrients such as energy, protein, and  vitamin D. That is, even when nutrients are consumed in the form of food, they may not  be properly absorbed if masticatory performance is poor” cannot be accepted.

Reviewer 2 Report

Comments and Suggestions for Authors

In the cross-sectional study the authors aimed to examine relationship between masticatory performance, nutritional intake and frailty in 182 older participants, mean age of 70 years.  They found that the robust group had significantly higher protein–energy ratio, vitamin D intake, subjective and objective masticatory performance than the frail group. Logistic regression analysis revealed significant association of the skeletal muscle mass, protein-energy ratio, and objective masticatory performance with frailty.

The main comment refers to the interpretation of the logistic regression results, according to which the authors conclude on association between frailty and masticatory performance, independent of nutrition.

Comments:

Line 116: Please,  specify what "This" means? What was measured?

Line 153: Specify what method InBody devices use for body composition measurement.

Line 162: Please, specify the reference category in dependent variable.

 Table 1, second column: It would be more appropriate to write "older" and "younger" because we cannot really consider respondents younger than 79 years to be “young”.

 Table 2: State in the title what is the dependent variable.

 Line 205: You should perform a multicollinearity test between the independent variables to conclude that the masticatory performance is related to frailty independent of nutrition. It is highly probable that there is a association between masticatory performance and nutrition, especially energy intake, as independent variables. Therefore,  the conclusion about the connection between frailty and masticatory performance - independent of nutrition, needs to be confirmed.

Line 242: Please see the previous comment.

Reviewer 3 Report

Comments and Suggestions for Authors

1.     Lines 207-214, …exceeded, lower, within…., Hence,     the readers cannot follow the authors’ pace. Please add the real value data to help readers catch the meaning.

2.     Lines 230-234, the writing is not very clear. It needs to clarify.

3.     Lines 239-255, this paragraph needs more deliberate discourse.

4.     Please check the meaning between lines 286-290 and lines 239-255.

Round 2

Reviewer 1 Report

Comments and Suggestions for Authors

The corrections made allow the article to be published